# 3D Structures of IgA, IgM, and Components

**DOI:** 10.3390/ijms222312776

**Published:** 2021-11-26

**Authors:** Shunli Pan, Noriyoshi Manabe, Yoshiki Yamaguchi

**Affiliations:** Division of Structural Glycobiology, Institute of Molecular Biomembrane and Glycobiology, Tohoku Medical and Pharmaceutical University, 4-4-1 Komatsushima, Aoba-ku, Sendai 981-8558, Miyagi, Japan; 22052501@is.tohoku-mpu.ac.jp (S.P.); manabe@tohoku-mpu.ac.jp (N.M.)

**Keywords:** IgA, IgM, J-chain, secretory component, glycosylation, molecular assembly, disulfide bridge, 3D structure

## Abstract

Immunoglobulin G (IgG) is currently the most studied immunoglobin class and is frequently used in antibody therapeutics in which its beneficial effector functions are exploited. IgG is composed of two heavy chains and two light chains, forming the basic antibody monomeric unit. In contrast, immunoglobulin A (IgA) and immunoglobulin M (IgM) are usually assembled into dimers or pentamers with the contribution of joining (J)-chains, which bind to the secretory component (SC) of the polymeric Ig receptor (pIgR) and are transported to the mucosal surface. IgA and IgM play a pivotal role in various immune responses, especially in mucosal immunity. Due to their structural complexity, 3D structural study of these molecules at atomic scale has been slow. With the emergence of cryo-EM and X-ray crystallographic techniques and the growing interest in the structure-function relationships of IgA and IgM, atomic-scale structural information on IgA-Fc and IgM-Fc has been accumulating. Here, we examine the 3D structures of IgA and IgM, including the J-chain and SC. Disulfide bridging and *N*-glycosylation on these molecules are also summarized. With the increasing information of structure–function relationships, IgA- and IgM-based monoclonal antibodies will be an effective option in the therapeutic field.

## 1. Introduction

Immunoglobulins (Igs) are categorized into several classes (IgM, IgD, IgA, IgG, and IgE), and all play an invaluable role in the immune response to pathogens. Natural IgG is a monomer composed of two heavy chains and two light chains. This simple structure allows relatively easy expression and purification. IgG is the most studied immunoglobin class and is frequently used in antibody therapeutics. In contrast, IgA was initially considered to be a non-inflammatory antibody for the maintenance of mucosal homeostasis [1]. It has since been found that the different assembly forms of IgA and IgM and their interaction with different receptors allow them to actively or passively inhibit or initiate inflammatory responses, thus forming the basis of mucosal immunity [2,3,4]. Unlike IgG, IgA and IgM are assembled into dimers or pentamers in the presence of J-chains. Moreover, they are transported to the mucosal surfaces by a secretory component (SC) as the first-line defense against infections [5]. Due to this structural complexity and inherent flexibility, 3D structural study at atomic scale has only recently commenced. Until now, structure–function relationships could only be discussed in crude terms based on low-resolution images and homology models. Without detailed knowledge of the 3D atomic structure and species differences, the development of this class of antibody therapeutics has been impeded. This situation is improving rapidly, and now atomic-scale structural information on IgA and IgM molecules is available due to the use of cryogenic electron microscopy (cryo-EM) and X-ray crystallographic techniques, which reveal the mode of molecular assembly. Electron densities originating from attached *N*-linked glycans have been partially solved, and these should help gain a greater understanding of the role of *N*-glycan modifications in the functions of IgM and IgA.

## 2. Structure of IgA

IgA plays an important role in mucosal immunity and is the most abundant immunoglobulin, with even more being produced than the sum of the other immunoglobulins [6]. Unlike other human immunoglobulins, IgA exists either as a monomer or as soluble polymers. Only polymeric IgA can bind to polymeric immunoglobulin receptors (pIgR) for transcytosis, with the presence of J-chain being required for binding [7,8,9]. In common with other immunoglobulins, the IgA monomer consists of two identical heavy chains and two identical light chains forming a Y-shape structure (Figure 1). Each chain extends a variable region from its N-terminal, followed by a constant region. The heavy chains of IgA are composed of four domains (V_H_, C_α_1, C_α_2, and C_α_3), while the light chains are divided into two domains (V_L_ and C_L_) [10]. The hinge region of IgA lies between the C_α_1 and C_α_2 structural domains of each heavy chain. The hinge affords the flexibility to the whole IgA molecule. This is critical for its activity, and it is this region that differs most between the two subclasses IgA1 and IgA2 (being particularly extensive in human IgA1 but absent in human IgA2). Unlike IgG, there is an 18-residue tailpiece extension at the C-terminus of the IgA and IgM heavy chains, which is essential for immunoglobulin assembly [11]. Fab is composed of V_H_, C_α_1, V_L_, and C_L_ domains and engaged in antigen binding. Fc comprises two C_α_2 and two C_α_3 domains, responsible for triggering effector functions through interaction with complements or Fc_α_ receptors [3]. Fc is stabilized by several intra-molecular disulfide bonds, Cys266-Cys323 (C_α_2), Cys369-Cys432 (C_α_3), and every Cys242 forms an inter-chain disulfide bond with the Cys299 of another chain (Figure 1). The major differences between IgA1 and IgA2 are the hinge region (an octapeptide sequence Pro-Ser-Thr-Pro-Pro-Thr-Pro-Ser is repeated in IgA1) and the number and location of *O*- and *N*-glycans [7,12,13]. The Cys133 of IgA1 C_α_1 forms a disulfide bond with the light chain, which is absent in IgA2 [14]. There are three IgA2 allotypic variants: m1, m2, and mn. These allotypes differ in disulfide bridging between a heavy chain and light chain so that Cys220 forms a covalent linkage when followed by Arg221 (IgA2 m2 and IgA2 mn) but not by Pro221 (IgA2 m1). In IgA2 m1, the replacement of Arg221 with Pro221 introduces a kink in the peptide backbone, which precludes the formation of heavy chain-light chain disulfide bridges but instead induces the formation of light chain-light chain disulfide bonds [13,14,15,16]. In contrast to the previous speculation that IgA2 mn could form heavy chain-light chain disulfide bonds in the absence of Cys220, mutation of Cys241 or Cys242 interferes with such disulfide bond formation [17]. Serum IgA molecules are mostly monomeric, while secretory IgA molecules are found as dimers and, to a lesser extent, as trimers and tetramers. In the dimer, two IgA molecules are linked together by another polypeptide designated as J-chain. Dimeric IgA is bound by the pIgR on the basolateral surface of epithelial cells, followed by the secretory component (SC) of the receptor covalently binding to dIgA to form secretory IgA (sIgA) [18,19].

Crystallization is often difficult due to the inherent flexibility of intact IgA monomers. Earlier electron microscopy (EM) studies showed that sIgA has a double-Y-like shape [20]. With the significant advances in cryo-EM in recent years, three-dimensional structures of sIgA-Fc dimers have been determined [21]. The two Fc_α_ are arranged in a boomerang-like shape with four tailpiece β-strands bundled together to mediate the interactions between the two Fc_α_ molecules (Figure 2) [21,22]. The dIgA is further stabilized through disulfide bridges between the Fc tail (Cys471) and the J-chain (Cys15 or Cys69), in which two parallel β-strands of Fc1 and Fc2 extend to the bottom and top sheets of the twisted J-chain β-sandwich. In the sIgA structure, interactions between SC domains and dIgA (disulfide bonding between Cys468 of SC and Cys311 in the C_α_2 domain) stabilize a bent and tilted relationship (an angle of 110° caudally) between two IgA monomers. Two or three additional Fcs are added to the plane of the original dimer to form a higher-order multimer [21,22,23,24,25].

Amino acid sequences of IgA show significant species differences in mammals and birds; however, *N*-linked glycosylation sites of the tailpiece are well conserved [26]. Both isotypes of IgA have two conserved *N*-linked glycosylation sites, Asn263 in C_α_2 and Asn459 in the tail piece. Asn263 has mostly biantennary complex-type glycan with α2-6-linked sialic acids, while Asn459 is dominated by triantennary glycan with α2-6 and α2-3-linked sialic acids [27,28]. *N*-linked glycans at Asn263 in C_α_2 are located on the outer surface of Fc to avoid potential instability caused by considerable surface exposure of the unpaired C_α_2 structural domain [23,29]. Recent studies suggest that *N*-glycan attached on Asn263 does not impact IgA binding to Fc_α_R1 [30], consistent with earlier reports [31,32]. Asn459 in the IgA2 tail piece has complex- and high mannose-type glycans but is not fully glycosylated [33]. The glycan controls the formation of IgA polymers, and sialic acid residues found in the complex *N*-linked glycan mediate antiviral activity [30,33]. The sialic acid residues directly interact with certain viruses, seemingly acting as bait to neutralize them [30]. Human IgA2 has additional *N*-glycosylation sites (Asn166 in the C_α_1 and Asn337 in the C_α_2), and IgA2m(2) has an additional *N*-linked glycan at Asn211 in the C_α_1 domain. These additional sites attach more neutralizing glycans, which creates a greater proinflammatory response of neutrophils and macrophages [31,34,35,36]. However, details of the mechanism are not known.

## 3. Structure of IgM

Immunoglobulin M (IgM) is another abundant antibody subclass active in human mucosal immunity, providing the first line of defense against pathogens [18,37]. The predominant form of mouse and human IgM antibodies is pentameric, but there are also small amounts of hexamers and monomers [38]. The multiple antigen-binding sites give IgM a higher apparent affinity for antigen. Each IgM monomer is composed of two heavy chains with five domains (V_H_, C_μ_1, C_μ_2, C_μ_3, and C_μ_4) and two light chains with two domains (V_L_, C_L_). The heavy chains are covalently linked with an inter-chain disulfide bond Cys337–Cys337, and each light chain is bonded to Cys136 in the heavy chain (Figure 3). Every Fc is stabilized with two intra-molecular disulfide bonds: Cys367–Cys426 (C_μ_3) and Cys474–Cys536 (C_μ_4). According to the sequence alignment, C_μ_1, C_μ_3, and C_μ_4 of IgM are equivalent to the C_γ_1, C_γ_2, and C_γ_3 of IgG. C_μ_2 of IgM is an additional constant domain that corresponds to the hinge region of IgG, providing the flexibility necessary for binding to antigens on cell surfaces. Cys414 of the C_μ_3 domain is essential for IgM polymerization and can form an inter-monomeric disulfide bond with the Cys414 of an adjacent monomer. Like IgA, there is a tailpiece with a short 18 amino acid peptide sequence at the C-terminus of the heavy chain. The Cys575–Cys575 disulfide bond in the tailpieces is critical for IgM polymerization. Five IgM monomers are linked by disulfide bonds to each other and to the J-chain to form a pentamer, while the J-chain is absent in the IgM hexamer [38,39,40,41,42,43].

The molecular size of a fully assembled IgM complex is nearly a million Daltons; hence, determination of its detailed structure is challenging. Earlier studies suggested that the pentamer in the presence of the J-chain has a star-like appearance with five-fold symmetry [44,45]. However, a recent cryo-EM study proves that five Fc_μ_ units are arranged in almost perfect hexagonal symmetry with a 61° gap occupied by the J-chain, and a triangular SC perpendicularly docked to the Fc_μ_-J near-planar structure to form secretory IgM (sIgM) [46] (Figure 4). C_μ_3–C_μ_4 and the tailpiece play important roles in the formation of IgM pentamers. Two neighboring C_μ_3–Cys414 form interchain disulfide bonds, and two neighboring IgM-C_μ_4 domains interact with each other by their FG loops (a loop between F and G strands). Ten tailpieces are arranged into two five-stranded parallel β-sheets. The two β-sheets stack together in an antiparallel fashion, which is critical for IgM pentamer stability. Cys575 residues of two adjacent Fc_μ_ form a disulfide bond and Cys575 on the side of the hexagonal gap is linked to the J-chain, which is essential for IgM oligomerization [44,46,47].

There are five conserved *N*-glycosylation sites: Asn171 (C_μ_1), Asn332 (C_μ_2), Asn395 (C_μ_3), Asn402 (C_μ_3), and Asn563 (tailpiece): Asn171, Asn332, and Asn395 are exposed to solvent with complex-type glycans (differences in glycan branching and sialylation) attached. These glycans are vulnerable to enzymatic processing and involved in the binding of IgM to T-cell surface receptors (asialo IgM remains bound, while sialylated IgM is internalized) [48]. The glycosylation of Asn402 (as with Asn297 of IgG) is crucial for complement activation with high mannose-type glycans. Asn563 of the tailpiece, located in the center of the IgM polymer, possesses high-mannose-type glycans which facilitate assembly of the IgM monomer and are necessary to prevent IgM aggregation [49,50].

## 4. Structure of J-chain

The J-chain was identified in 1971 in fractions from sIgA and sIgM, but not in those from IgG, IgE, and IgD [51]. It is highly conserved among different species, and there is no other homologous protein [52]. Human J-chain is a 15-kDa acidic polypeptide composed of 137 amino acids, including six Cys residues involved in intramolecular disulfide bridges (Cys12–Cys100, Cys71–Cys91, and Cys108–Cys133) and two in intermolecular (Cys15 and Cys69) disulfides to the Fc of IgM and IgA (Figure 5) [53]. Although various predictions of the structure of the J-chain have been made based on the pattern of disulfide bonds, its structure has not been resolved until recently. Now, with the structures of sIgA and sIgM determined, the three-dimensional structure of the J-chain is resolved in their polymers. As predicted, the J-chain consists almost entirely of β-sheets and loops in the IgA dimer and the IgM pentamer [21,46,53]. In the IgA dimer, the central region of the J-chain contains four β-strands (β1–β4) and three hairpins (Figure 5). β1–β3 strands are assembled into a β-sheet together with two tailpiece strands of Fc_α_2. Strand β4 is packed onto the tailpieces of Fc_α_1 to form another β-sheet. These two β-sheets fold into a single β-sandwich-like domain through strong hydrophobic interactions. β-hairpins 1 and 2 interact with the top surface of Fc2, while the bottom surface of Fc1 interacts with β-hairpin 3. β-hairpin 1, β3–β4 loop, β-hairpin 2, and C-terminal β-hairpin 3 form four lassos to further interact with the Fc_α_. Cys69 and Cys15 of the J-chain form disulfide bonds with the penultimate cysteine residues Cys471 of Fc_α_1 and Fc_α_2, which are the foundation for the locking of two IgA monomers to the J-chain [21,22,25]. As in the IgA dimer, the J-chain forms a radially centered β sandwich-like structure with the tailpieces of five Fcs in the IgM pentamer. The β3 and β4 strands are packed on the tailpieces of Fc5 and Fc1 through a disulfide bond Cys15 (J-chain)-Cys575 (Fc1), while the β2–β3 loop interacts with the base of Fc5 through a disulfide bond C69 (J-chain)-Cys575 (Fc5). The hairpin of the C-terminal wing forms extensive hydrophobic interactions with Fc1 and also interplays with pIgR/SC, allowing the IgM to attach to the receptor [46]. The biggest difference between the J-chains of the sIgA and sIgM structures is that β-hairpin 2 in the IgA dimer is not observed in IgM [54]. The structure of the J-chain without IgA/IgM has not been reported yet. There is a conserved *N*-glycosylation site at Asn49 of the human J-chain. The attached oligosaccharides are represented in three forms, each differing in the amount of sialic acid, 30% contain two sialic acid residues, 55% contain a sialic acid residue on the α l,3-linked mannose branch, and 15% have no sialic acid but two terminal galactose residues [55]. The glycosylated murine J-chain mutant with an Asn48 (corresponding to Asn49 of human J-chain) to Ala substitution in human IgA1 has reduced dimer assembly [56]. The *N*-glycosylation site of the J-chain is distant from any SC-pIgA interaction interface in sIgA, which may be relevant for promoting host and pathogen lectin binding [15,22,25]. Likewise, the glycan attached Asn49 is completely exposed to the surface of the human sIgM molecule without any contact with protein [46].

## 5. Structure of Secretory Component

Human pIgR plays a dominant role in mucosal immunity. The extracellular ligand-binding portion (known as SC) of pIgR is proteolytically cleaved and transported into mucosal secretions along with dimeric IgA (dIgA) or pentameric IgM (pIgM) [5]. SCs bind and exclude pathogens as part of the immune response [5,57,58]. Human SC contains 620 amino acids, forming five Ig-like domains named D1-D5 from the N-terminus [59]. There is a short non-immunoglobulin-like sequence between the D5 terminus and the membrane [60]. The three-dimensional structure of SC in a free form has been determined by X-ray crystallography, and the five domains (D1–D5) are organized into a compact, plate-like isosceles triangle (Figure 6). D2–D3 and D4–D5 form two edges, and D1 comes into contact with D2 and D4–D5 to form a third edge [19]. There are seven *N*-glycosylation sites (Asn65, 72, 117, 168, 403, 451, and 481), with fucosylated and sialylated *N*-glycans [61]. The ordered glycans are observed at four *N*-linked glycosylation sites (Asn65, Asn72, Asn168, and Asn481) and are mostly exposed on surfaces of the molecule. [19,60] They help the SC bind to pIg ligands and facilitate interactions between lectin and free SC, in addition to protecting SC from proteolytic degradation [5,22]. Each domain has a basic folded topology consisting of two β-sheets connected by a conserved disulfide bond (Cys22–Cys92 in D1, Cys134–Cys202 in D2, Cys239–Cys307 in D3, Cys353–Cys423 in D4, and Cys468–Cys526 in D5) [19,60,61]. With the determination of the sIgA and sIgM structures, the 3D structure of SC bound to IgA and IgM was also reported. A drastic conformational change of SC is observed in sIgA and sIgM (Figure 6). The D4–D5 side rotates 120° to be colinear with unchanged D2–D3, and D1 is packed on D2–D3 by rotating 84° to allow binding to dIgA and pIgM [21,46]. Noncovalent interactions with three complementary determinants (CDRs) of D1 play an essential role in the assembly of sIgA or sIgM in the presence of the J-chain, and there is a broadly consistent D1 interaction interface in sIgA and sIgM. D2–D4 provide the correct spacing of D1 and D5, thereby assisting interaction with dIgA and pIgM [21,25,46,54]. D5 interacts with Fc through the disulfide bond between Cys468 (SC D5) and Cys311 (IgA Fc2) or Cys414 (IgM Fc2) [62,63].

## 6. Interaction with Fc Receptors and Others

IgA and IgM Fcs are known to interact with a variety of receptors to mediate multiple effector functions, such as pIgR, Fc_α_/_μ_R, Fc_α_RI, complement receptor, Fc_μ_R, etc. Under homeostatic conditions, IgA complexed with pIgR inhibits bacterial adherence to the mucosal wall, which functions as the first line of defense of the body. Interactions with pIgR have been discussed in the SC section. Once pathogens enter the bloodstream, serum monomeric IgA initiates the third line of immune defense as a proinflammatory antibody interacting with the IgA-specific receptor Fc_α_RI. Here, we elaborate on the interactions with human Fc_α_/_μ_R and the IgA-specific receptor Fc_α_RI, and IgM with the apoptosis inhibitor of macrophages (AIM). IgA interacts with Fc_α_RI to trigger clearance of IgA-bound aggressive pathogens. Most IgA-dependent immune responses are mediated through Fc_α_RI, inducing phagocytosis, antibody-dependent cell-mediated cytotoxicity, respiratory burst, and cytokine release [64]. Crystal structures of the Fc_α_RI complex with IgA1 Fc have greatly facilitated our understanding of their interactions with IgA. Like most Ig receptors, Fc_α_RI is a transmembrane glycoprotein comprising two extracellular Ig-like structural domains (EC1 and EC2), a transmembrane region, and a short cytoplasmic tail. The two extracellular structural domains are folded together at an angle of approximately 90° [23,32,65]. In each extracellular domain, several β-strands are assembled into two β-sheets, where the first β-strand is shared between the two β-sheets and with one strand absent in EC2. Both domains include three short helices, and there is an extra polyproline type-II helix in EC2 [23]. There are six potential *N*-glycosylation sites (Asn44, Asn58, Asn120, Asn156, Asn165, and Asn177) and several *O*-glycosylation sites in free Fc_α_RI. A recent study shows that Asn177 may not be glycosylated [66,67]. IgA monomer and polymer can all bind Fc_α_RI. The crystal structure of the Fc_α_RI/Fc_α_ complex shows that a single Fc dimer binds two Fc_α_RI receptors. The interaction between Fc_α_RI and IgA occurs at the C_α_2-C_α_3 interface, consisting of a central hydrophobic core and two sides of charged residues [23,66,68,69]. Free Fc_α_RI receptor binds to Fc_α_ with a conformational change in two loops and a strand in EC1 [23]. Asn58 of Fc_α_RI is closest to the IgA interaction surface, and deglycosylation at Asn58 results in increased IgA binding [67]. There is an increase in affinity upon desialylation at Asn58, and Asn58 modified with a single GlcNAc residue has the highest receptor affinity [70]. The glycans attached at the other sites exert little or no influence on IgA binding [67,70].

Another novel Fc receptor Fc_α_/_μ_R, the only IgM Fc receptor on hematopoietic cells, plays a significant role in triggering IgA- and IgM-mediated immune responses [71,72,73]. The Fc_α_/_μ_R gene is located adjacent to pIgR on chromosome 1. Amino acid sequence alignment shows that there is a conserved motif in SC D1 that is crucial for interacting with sIgA and pIgM [74,75]. This type-1 transmembrane protein has a unique extracellular loop containing three domains, a short EC1, an Ig variable region-like EC2 that shares 43% homology with SC D1, and a stalk-like EC3 [76]. Studies using an antibody against Fc_α_/_μ_R and 3D homology modeling suggest the presence of a conserved peptide sequence (also present in SC D1) that interacts with IgA or IgM [73,77,78]. Like pIgR, Fc_α_/_μ_R has three CDR-like loops in EC2, and the CDR1-like loop lies exactly at the position of the conserved peptide sequence (TIHCHYAPSSVNRHQRKYW). Substitution experiments indicate that the CDR1- and CDR2-like loops together play a more essential role than the CDR3-like loop in binding to IgA and IgM, where positively charged Arg31 and the hydrophobic side chain of Val29 of the CDR1-like loop and Asn54-Gln55 of the CDR2-like loop contribute to ligand binding, while the CDR3-like loop assists in stabilizing the structure. Fc_α_/_μ_R has a higher affinity for IgM than IgA and only binds to IgM and IgA polymers [78,79,80]. There are two potential *N*-glycosylation sites at Asn167 and Asn276, but the function of putative *N*-glycans in binding to IgA and IgM is unclear [74,81].

Apoptosis inhibitor of macrophages (AIM) is a member of a scavenger receptor cysteine-rich (SRCR) superfamily and is expressed on the surface of macrophages [82]. Recent studies show that the pentameric IgM is a transporter for the effector protein AIM and that the trapping by pIgM inactivates AIM and increases its level in the blood by decreasing renal excretion [83,84]. AIM is released from an AIM–IgM complex to inhibit the apoptosis of thymocytes, especially during acute kidney injury, and this enhances the clearance of excess fat, bacteria, cancer cells, and dead cell debris [85,86]. sIgM is able to bind to the complement receptor and Fc_α_/_μ_R on antigen-presenting cells, including macrophages and follicular dendritic cells (FDC) [87]. AIM-containing IgM-antigen complexes are ligated to the FDC surface by complement receptors. In contrast, cells lacking AIM bind to Fc_α_/_μ_R and are processed and present on MHC class II after internalization [88].

Single-particle negative-stain EM of a pIgM-AIM complex has shown how pIgM associates with AIM [47]. There are three conserved SRCR domains in the AIM molecule. SRCR2 and SRCR3 are essential for the formation of the pIgM-AIM complex, especially Cys194 of SRCR2. SRCR1 does not appear to bind pIgM according to a homology-based structural model and mutational analysis. AIM assumes a broad bean-like structure, located in a 50° gap in pIgM with the J-chain. The Cys194 of SRCR2 forms an intermolecular disulfide bond with the Cys414 of the Fc-C_μ_3 domain at one edge, and the positively charged amino acid cluster (H294, K298, R300, K301, K340) of SRCR3 interacts with the negatively charged cluster of the Fc-C_μ_4 domain on the opposite edge [47,89,90]. There are three potential *N*-glycosylation sites (Asn99 of SRCR1, Asn229 of SRCR2, and Asn316 of SRCR3) in murine AIM and these *N*-glycans affect the secretion and lipolytic functions. It is unclear whether the *N*-glycans play a role in IgM interactions [91]. Potential *N*-glycosylation sites of human AIM do not appear to possess any glycans.

## 7. Therapeutic Potential of IgA and IgM

Although IgA and IgM play an essential role in human mucosal immunity, conventional vaccinations usually do not induce mucosal immunity unless administrated to the mucosal surface. The progress in determining their structure and interaction with receptors suggests that IgA and IgM-based immunotherapy is an important field for the development of vaccinations and therapeutics. More than 90 therapeutic IgG antibody products have FDA approval, whereas almost no IgA and IgM antibodies have been tested in humans [92]. The superiority of IgA over IgG for recruiting polymorphonuclear cells and the development of bispecific antibodies (BsAb) have opened up prospects for IgA antibody applications [93,94,95]. For instance, anti-HER2 × Fc_α_RI (against both HER2 and Fc_α_RI) effectively clears breast cancer cells through neutrophil accumulation, but anti-HER2 × Fc_γ_RI does not [96]. Another BsAb, anti-HLA II × Fc_α_RI is also a powerful candidate in recruiting polymorphonuclear cells against human B cell malignancies [97]. In addition, bispecific IgM with ten or twelve binding sites potentially allows very high affinity binding to difficult or rare tumor antigens, enabling selective contact with T cells resulting in tumor cell death. Anti-CD20 × CD3 IgM is currently in clinical trials against refractory or resistant non-Hodgkin’s lymphoma [98,99].

While there are many possible advantages of IgA and IgM in antibody therapy, there are several issues that need to be overcome. For example, how does IgA mediate both pro- and anti-inflammatory functions via FcαRI in order to exert a therapeutic effect. The short half-life makes use of IgA potentially expensive, and the need for the high frequency of administration is inconvenient for patients. Another issue is the efficiency of expression, production, purification, and complete assembly of recombinant IgA and IgM monoclonal antibodies with appropriate homogeneity. A more critical limitation is the lack of suitable animal models, one example being the differences in the polymerization state of serum IgA and the lack of Fc_α_RI in the mouse. Many of these issues are now being addressed. For instance, modification of terminal *N*-glycans successfully lengthened the half-life of asialoglycoprotein receptor-mediated clearance [100,101]. Systems for increasing Igs expression and rational assembly are being gradually developed [33,101,102]. Successful establishment of human CD89 and IgA transgenic mice now enables in vivo studies of IgA [103,104].

The recently resolved three-dimensional structures of IgM Fc pentamers and IgA Fc multimers have provided us with an excellent base to help understand and modify these complex macromolecules. We hope that this review provides some ideas for the engineering of bispecific IgA and IgM antibodies and the development of more effective therapeutics.

## Figures and Tables

**Figure 1 ijms-22-12776-f001:**
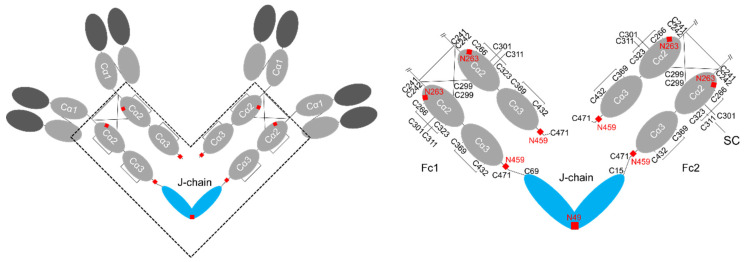
Schematic drawings of human IgA. Disulfide bonding and *N*-glycosylation sites are indicated.

**Figure 2 ijms-22-12776-f002:**
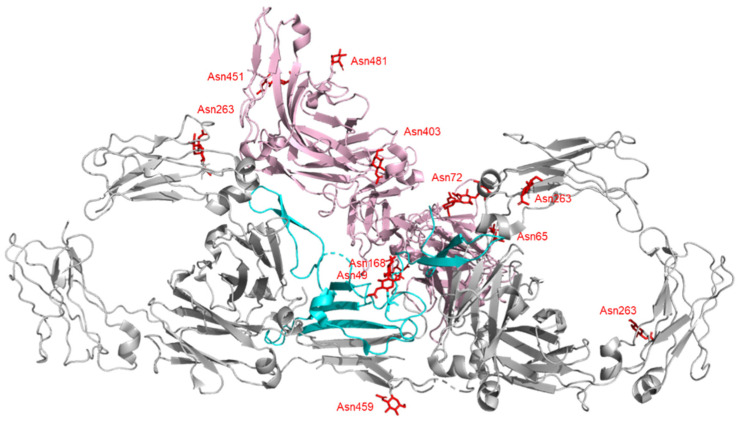
3D structure of human IgA-Fc dimer (gray) in complex with J-chain (sky blue) and SC (pink) determined by cryo-EM analysis (PDB ID: 6UE7). Glycan residues are shown in red with stick representation.

**Figure 3 ijms-22-12776-f003:**
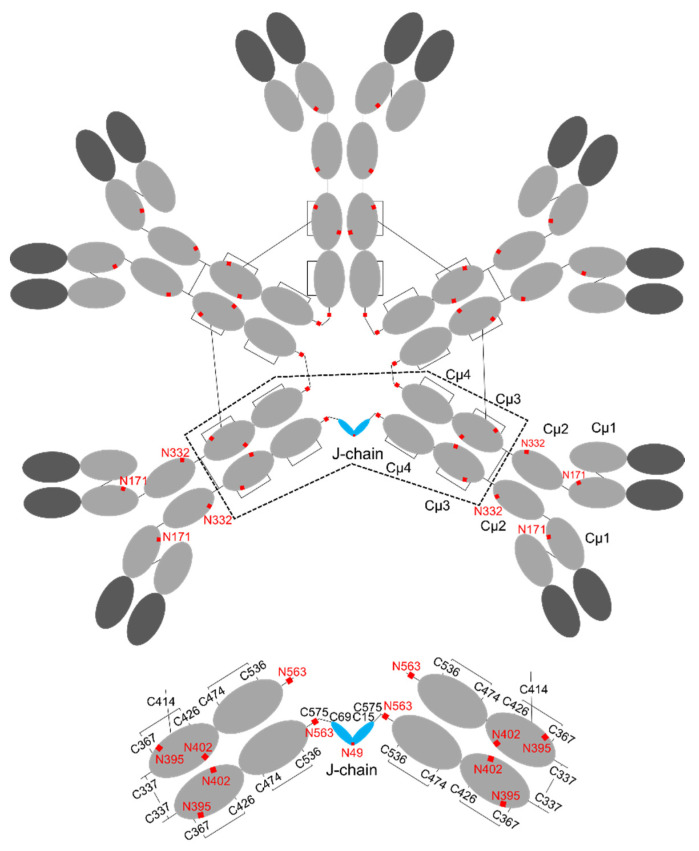
Schematic drawings of human IgM pentamer. Disulfide bonding and *N*-glycosylation sites are indicated.

**Figure 4 ijms-22-12776-f004:**
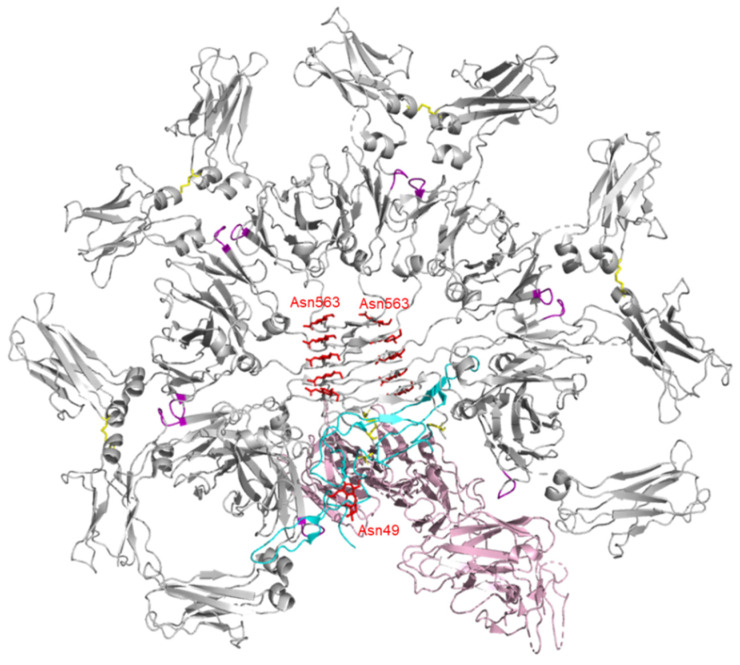
3D structure of human IgM-Fc pentamer (gray) in complex with J-chain (sky blue) and SC (pink) determined by cryo-EM analysis (PDB ID: 6KXS). Cys414 disulfide bonds and FG loops in Fc are highlighted in yellow and magenta, respectively. Glycan residues are shown in red with stick representation.

**Figure 5 ijms-22-12776-f005:**
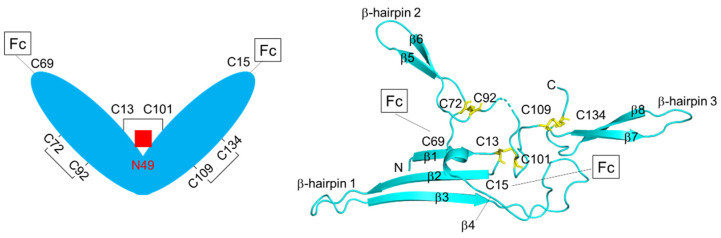
Schematic drawings of human J-chain. Disulfide bonding and *N*-glycosylation site are indicated. 3D structure of human J-chain bound to IgA-Fc is shown on the right (PDB ID: 6UE7). IgA-Fc and SC are omitted for clarity.

**Figure 6 ijms-22-12776-f006:**
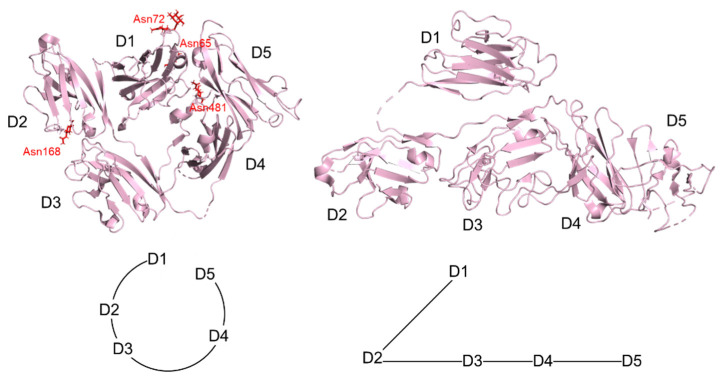
3D structures of SC in free form (PDB ID: 5D4K, (**left**)) and SC bound to IgA-Fc dimer and J-chain (PDB ID: 6UE7, (**right**)). Glycan residues are shown in red with stick representation. On the right panel, IgA-Fc was omitted for clarity.

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
