# Peer review of "3D Structures of IgA, IgM, and Components"

_ijms, 2021, doi:10.3390/ijms222312776_

Round 1
Reviewer 1 Report
This review describes about 3D structures of IgA and IgM, including the J-chain and secretory component. In addition, disulfide bridging and N-glycosylation are summarized as well as therapeutic potential of IgA and IgM. This manuscript is organised well and written well. It is recommended to be acceptable after minor revision.
- Abs, Impact of the issue can be provided at the end of abstract.
- Abs, at the end part, secretory component -> SC
- Figures 1 and 3, font size should be enlarged.
Author Response
Abs, Impact of the issue can be provided at the end of abstract.
-> At the end of abstract, we add a sentence “With the increasing information of structure-function relationships, IgA- and IgM-based mono-clonal antibodies will be an effective option in the therapeutic field.”
Abs, at the end part, secretory component -> SC
-> We changed secretory component into SC.
Figures 1 and 3, font size should be enlarged.
-> We enlarged the font size of Figures 1 and 3, thank you.
Reviewer 2 Report
The authors present a review on the recent developments in the 3D structure study of IgA and IgM immunoglobulins. Compared to IgG, the most well-studied antibody type, IgA and IgM have received less attention, primarily due to their more complex nature and organization into multimers/polymers. As such, the recent determination of IgA and IgM structures through X-ray crystallography and cryo-EM constitute important findings in antibody research.
The review is written in very good English, with very few syntax errors (which can be easily corrected in the proofing stage). The structural aspects of IgA and IgM, their unique features compared to IgG, and the characteristics of their subtypes are described adequately, both in the text and in the accompanying figures. In addition, the biological and biomedical significance of the two immunoglobulin types are presented in a comprehensive manner.
I therefore recommend that this manuscript be ACCEPTED for publication.
Author Response
Thank you for comments. We will carefully check the syntax errors in the proofreading state.